# Use of non-invasive intracranial pressure pulse waveform to monitor patients with End-Stage Renal Disease (ESRD)

Cristiane Rickli[1], Lais Daiene Cosmoski[1], Fábio André dos Santos[1], Gustavo Henrique Frigieri[2], Nicollas Nunes Rabelo[3], Adriana Menegat Schuinski[1], Sérgio Mascarenhas[2], José Carlos Rebuglio Vellosa[1] *

**1** Biological and Health Sciences Division, State University of Ponta Grossa–UEPG, Ponta Grossa-PR, Brazil, **2** Braincare Desenvolvimento e Inovação Tecnológica S.A., São Carlos-SP, Brazil, **3** Neurosurgery Department, Center University UniAtenas, Paracatu-MG, Brazil

\* josevellosa@yahoo.com.br

**Data Availability Statement:** All relevant data are within the manuscript and its Supporting Information files.

## Abstract

End-stage renal disease (ESRD) is treated mainly by hemodialysis, however, hemodialysis is associated with frequent complications, some of them involve the increased intracranial pressure. In this context, monitoring the intracranial pressure of these patients may lead to a better understanding of how intracranial pressure morphology varies with hemodialysis. This study aimed to follow-up patients with ESRD by monitoring intracranial pressure before and after hemodialysis sessions using a noninvasive method. We followed-up 42 patients with ESRD in hemodialysis, for six months. Noninvasive intracranial pressure monitoring data were obtained through analysis of intracranial pressure waveform morphology, this information was uploaded to Brain4care® cloud algorithm for analysis. The cloud automatically sends a report containing intracranial pressure parameters. In total, 4881 data points were collected during the six months of follow-up. The intracranial pressure parameters (time to peak and P2/P1 ratio) were significantly higher in predialysis when compared to postdialysis for the three weekly sessions and throughout the follow-up period ($p < 0.01$) data showed general improvement in brain compliance after the hemodialysis session. Furthermore, intracranial pressure parameters were significantly higher in the first weekly hemodialysis session ($p < 0.05$). In conclusion, there were significant differences between pre and postdialysis intracranial pressure in patients with ESRD on hemodialysis. Additionally, the pattern of the intracranial pressure alterations was consistent over time suggesting that hemodialysis can improve time to peak and P2/P1 ratio which may reflect in brain compliance.

## Introduction

Chronic kidney disease (CKD), a leading cause of mortality and morbidity and a growing public health problem worldwide [1], is a complex disease that requires multiple treatment approaches [2].

**Funding:** The Braincare Desenvolvimento e Inovação Tecnológica S.A. provided the equipment free of charge for this study. CR declares Scholarship Funding from CAPES - Coordenação de Aperfeiçoamento de Pessoal de Nível Superior – Finance Code 001. This financing did not role in study design, data collection and analysis, decision to publish, or preparation of the manuscript. GHF declares personal fees as employee (Research Coordinator) from Braincare Desenvolvimento e Inovação Tecnológica S.A., during the conduct of the study; In addition, GHF has a patent US9826934B2 issued, and a patent US9993170B1 issued. The funder (Braincare Desenvolvimento e Inovação Tecnológica S.A.) only provided support in the form of salaries for author GHF, but did not have any additional role in the study design, data collection and analysis, decision to publish, or preparation of the manuscript. GHF contributed as a researcher with Conceptualization, Methodology, Software, Visualization, Writing – review & editing, as described in the 'author contributions' section. This commercial affiliation does not alter our adherence to PLOS ONE policies on sharing data and materials. NNR declares personal fees as medical consultant from Braincare Desenvolvimento e Inovação Tecnológica S.A., during the conduct of the study. The funder (Braincare Desenvolvimento e Inovação Tecnológica S.A.) only provided support in the form of consultant fee for author NNR, but did not have any additional role in the study design, data collection and analysis, decision to publish, or preparation of the manuscript. NNR contributed as a researcher with Methodology, Visualization, Writing – review & editing, as described in the 'author contributions' section. This commercial affiliation does not alter our adherence to PLOS ONE policies on sharing data and materials.

**Competing interests:** CR declares no competing interests/ has nothing to disclose. LDC declares no competing interests/ has nothing to disclose. FAS declares no competing interests/ has nothing to disclose. GHF declares personal fees as employee (Research Coordinator) from Braincare Desenvolvimento e Inovação Tecnológica S.A., during the conduct of the study; In addition, GHF has a patent US9826934B2 issued, and a patent US9993170B1 issued. The funder (Braincare Desenvolvimento e Inovação Tecnológica S.A.) only provided support in the form of salaries for author GHF, but did not have any additional role in the study design, data collection and analysis, decision to publish, or preparation of the manuscript. This commercial affiliation does not alter our adherence to PLOS ONE policies on sharing data and materials. NNR declares personal

Hemodialysis (HD) has become the predominant renal replacement therapy (RRT) in the world [3]. However, HD is associated with frequent complications, including hypotension and muscle cramps, in addition to postdialysis complaints of headache, fatigue, and inability to concentrate, which may significantly affect patients' quality of life [4]. Mild signs and symptoms like headache, nausea, and muscle cramps are often attributed to volume depletion due to excessive ultrafiltration, but may represent a milder but not diagnosed form of dialysis disequilibrium syndrome (DDS) [5].

Even though maintenance HD has been a routine procedure for over 50 years, the exact mechanism of DDS remains poorly understood [6] and the syndrome manifests as neurologic symptoms and signs related to osmotic fluid shifts [5]. Cerebral edema and increased intracranial pressure (ICP) are the primary contributing factors to this syndrome and are the targets of therapy [6]. Neurologic manifestations progress sequentially as cerebral edema worsens and ICP rises and, if not promptly recognized and managed, can lead to coma and even death [7].

ICP monitoring could assist in the early diagnosis of DDS, but the methods in use are highly invasive, costly, and carry complication risks. However, a noninvasive method based on volumetric skull changes detected by a sensor has been developed [8]. This method allows for quick and safe access to ICP pulse waveform morphology, which is correlated with brain compliance [9].

This study aimed to follow-up patients with end-stage renal disease (ESRD) by monitoring ICP before and after HD sessions using a noninvasive method to assess ICP variations during HD treatment.

## Materials and methods

### Participants

This study was approved by Research Ethics Committee of the State University of Ponta Grossa/COEP-UEPG (process number: 1.834.627). It is a prospective longitudinal study of 42 patients aged ≥ 18 years with end-stage renal disease (ESRD) from a single RRT center who received HD periodically, three times per week with two one-day intervals and one two-day interval between sessions, for six months. HD session length ranged from three to four hours depending on the patient and his/her condition. The authors declare that they adhered to the Declaration of Helsinki. All participants received information about the study and provided written informed consent.

The clinical characteristics of participants including age, gender, underlying disease, start of treatment, and comorbidities were retrieved from the electronic medical records of the RRT center. The parameters mean arterial pressure (MAP) and interdialytic weight gain (IDWG) were obtained by consulting the notes of each hemodialysis session.

### Intracranial pressure (ICP) and brain compliance monitoring

In total, 4881 data points were collected during the six months of follow-up. The noninvasive ICP monitoring equipment was provided by Brain4care® (São Paulo, SP, Brazil). This noninvasive method was validated by comparison with the invasive ICP monitoring method [8,10].

Predialysis monitoring sessions were done before the HD session in a private room with the patient seated in a chair similar to the one used in the HD session while ICP was monitored for 1 to 3 min. The patient should remain still during signal acquisition and the same procedure was performed after the HD session. Fig 1 shows a flowchart of how this search was conducted.

Monro-Kellie doctrine states that the skull does not expand after the fontanels are closed. However, for the development of the ICP monitor, it was proven that the skulls, even those of

fees as medical consultant from Braincare Desenvolvimento e Inovação Tecnológica S.A., during the conduct of the study. The funder (Braincare Desenvolvimento e Inovação Tecnológica S.A.) only provided support in the form of consultant fee for author NNR, but did not have any additional role in the study design, data collection and analysis, decision to publish, or preparation of the manuscript. This commercial affiliation does not alter our adherence to PLOS ONE policies on sharing data and materials. AMS declares no competing interests/ has nothing to disclose. SM declares he has a patent US9826934B2 issued, and a patent US9993170B1 issued. These patents issued do not alter our adherence to PLOS ONE policies on sharing data and materials. JCRV declares no competing interests/ has nothing to disclose.

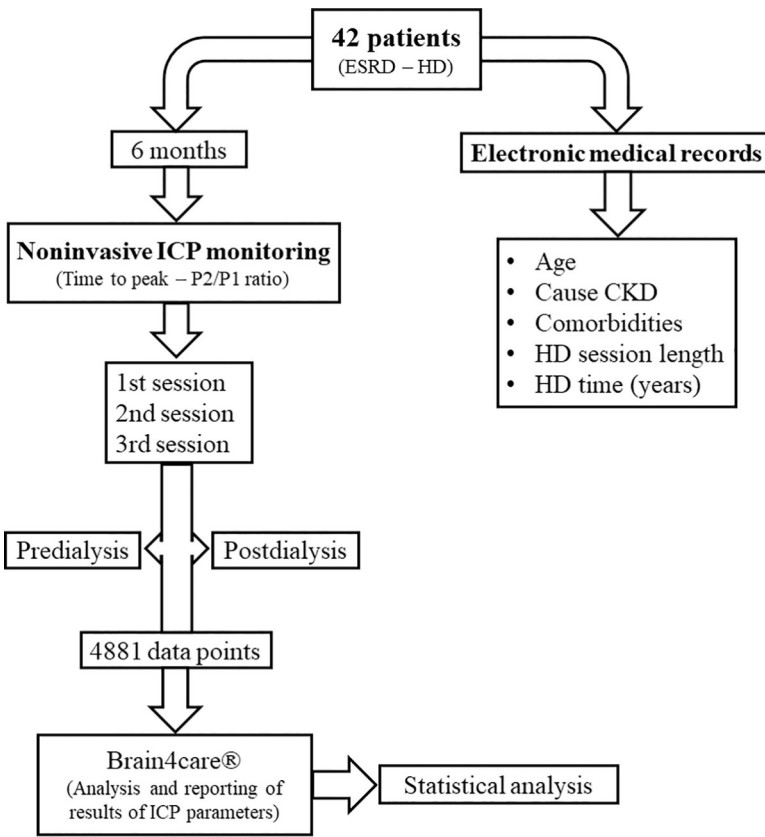

**Fig 1. Flowchart of this research.** CKD: Chronic kidney disease; ESRD: End-stage renal disease; HD: Hemodialysis; ICP: Intracranial pressure.

adults, have volumetric changes, as a result of pressure variations and the noninvasive monitoring of ICP is based on the assessment of these changes in bone structure [11].

To perform the monitoring, a sensor that detects the micrometric deformations of the skull bones is attached to a plastic headband strapped around the patient's head. The device filters, amplifies and digitizes the signal from the sensor before sending it to a computer [8].

Noninvasive ICP monitoring data were obtained through analysis of ICP waveform morphology. The ICP waveform is a modified blood pressure wave with three distinct peaks. The first peak (P1, or 'percussive wave') is the result of arterial pressure transmitted from the choroid plexus. The second peak (P2) signifies brain compliance and the last peak (P3) is the result of the aortic valve closure. Thus, under normal ICP conditions, the amplitude of the peaks is such that P1>P2>P3 [12,13]. However as brain compliance decreases and the ICP increases, the amplitude of the wave also increases and the P2 component of the wave exceeds P1 and P3 [14].

Following the ICP monitoring, the software saved the data to files that were later uploaded to Brain4care® for analysis. The result is a report containing the time to peak (TTP) and P2/P1 ratio. TTP was defined as the time at which the ICP curve reaches its tallest peak, either P1 or P2, starting from the start of the curve. The P2/P1 ratio assesses brain compliance and was defined as the ratio between the amplitudes of peaks P2 and P1 ($R = AmpP2/AmpP1$). Brain compliance is normal when $R < 1.0$ (P2<P1) and abnormal when $R > 1.0$ (P2>P1).

## Statistical analysis

Clinical characteristics are expressed as mean ± standard deviation (SD) for continuous variables and relative frequency (%) for categorical variables. First, the mean ± SD of noninvasive ICP parameters (TTP and P2/P1 ratio) were calculated separately for the three weekly HD sessions ($1^{st}$, $2^{nd}$, and $3^{rd}$) for each month of follow-up. The normality of the data (TTP and P2/P1 ratio) in each evaluation period at pre-dialysis and post-dialysis were assessed and confirmed with the Kolmogorov-Sminov test ($p > 0.05$).

Predialysis and postdialysis noninvasive ICP parameters were compared using the Student's t-test for paired samples. Besides, TTP and P2/P1 ratio were also compared between the three weekly HD sessions using repeated measures analysis of variance (ANOVA) and Tukey multiple comparison tests. For assessment of MAP and IDWG, volunteers were grouped and compared according to the dialysis session (1st, 2nd and 3rd) through analysis of variance (ANOVA) for repeated measures with Tukey's post-test. As for MAP, the paired Student's t test was used to compare the sessions. PAM and IDWG were correlated to TTP and the P2/P1 ratio through Person's correlation. A *p*-value of $<0.05$ was considered statistically significant. All analyses were performed using SPSS version $17.0^{®}$ for Windows (SPSS Inc., Chicago, IL, USA).

## Results and discussion

Table 1 shows the baseline characteristics of the study participants.

The results of the intracranial pressure (ICP) pulse waveform monitorings are shown in Fig 2. The parameters TTP and P2/P1 ratio were generally higher in the predialysis moment compared to the postdialysis moment. This change was repeated over time and a statistically significant difference was demonstrated in all evaluated sessions and months ($p < 0.01$, paired Student's t-test).

**Table 1. Clinical characteristics of the study patients\*.**

| Clinical parameter | Value |
|---|---|
| Age (years) | 55.8 ± 16.5 |
| Age ≥ 60 years (%) | 50.0 |
| Female gender (%) | 45.2 |
| CKD cause (%) | |
| Undetermined | 23.8 |
| Multifactorial | 11.9 |
| diabetic nephropathy | 11.9 |
| chronic glomerulonephritis | 11.9 |
| polycystic kidney disease | 9.5 |
| hypertensive nephrosclerosis | 4.8 |
| Other | 26.2 |
| Comorbidities (%) | |
| systemic arterial hypertension | 64.3 |
| diabetes mellitus | 21.4 |
| Mean HD session length (min) | 220.0 ± 23.9 |
| HD session length ≥ 240 min (%) | 52.4 |
| Mean HD time (years) | 4.8 ± 4.8 |

\*Sample size: 42 patients with CKD on regular hemodialysis.

CKD, chronic kidney disease; HD, hemodialysis.

Values are mean ± standard deviation or relative frequency (%).

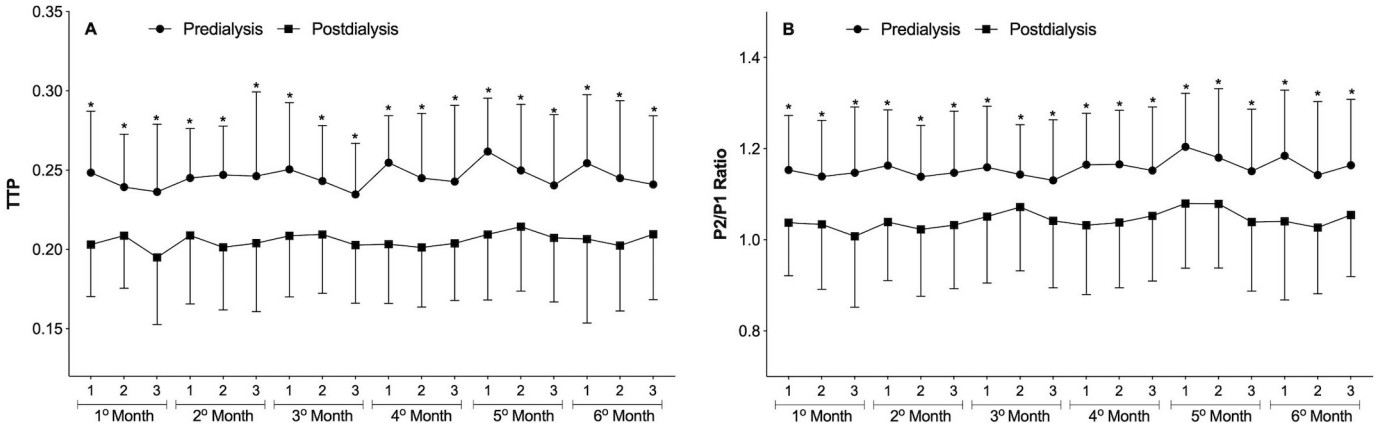

**Fig 2. Predialysis and postdialysis noninvasive intracranial pressure (ICP) parameters at the three weekly hemodialysis sessions over the six months of follow-up.** A) Time to peak (TTP). B) P2/P1 ratio. Differences between predialysis and postdialysis intracranial pressure parameters were analyzed by paired Student's t-test (*$p < 0.01$). Differences in TTP and P2/P1 ratio between the three weekly hemodialysis sessions were analyzed by repeated-measures ANOVA but were not significant ($p > 0.05$). Data are presented as Mean ± SD.

Still, in Fig 2, a second analysis comparing ICP parameters between the three weekly hemodialysis sessions over the six months of follow-up showed no significant differences in predialysis and postdialysis measurements. However, when all monitorings from each HD session (1st, 2nd, and 3rd weekly session) were pooled, there were significant differences in predialysis TTP and P2/P1 ratio with higher non-invasive ICP parameter values in the first session of the week (Fig 3) ($p < 0.05$, repeated-measures ANOVA followed by the Tukey's post-hoc test).

The main finding of this study was the significant difference detected between predialysis and postdialysis noninvasive ICP parameters (TTP and P2/P1 ratio) over the six months of follow-up (Fig 2). These parameters obtained through analysis of ICP waveform morphology. According to Nucci et al. (2016), changes in ICP waveform morphology can reflect changes in

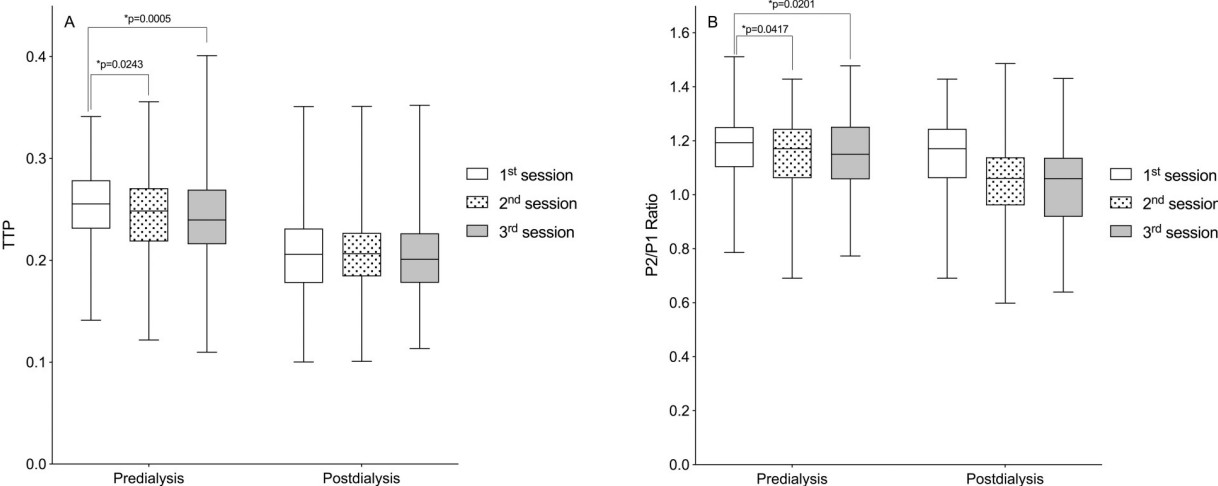

**Fig 3. Comparison of predialysis and postdialysis noninvasive intracranial pressure (ICP) parameters according to the hemodialysis sessions over the six months of follow-up.** (A) Time to peak (TTP). (B) P2/P1 ratio. Box-plots illustrate the median and interquartile range of the noninvasive intracranial pressure parameters at the three weekly hemodialysis sessions (1st, 2nd, and 3rd) over the six months of follow-up (*$p < 0.05$, repeated-measures ANOVA followed by the Tukey's post-hoc test).

ICP whereas ICP wave morphological analysis can in turn predict the ICP measurements of invasive methods [15].

Considering that in general the parameters of non-invasive ICP were higher in the predialysis moment, it is suggested that the removal of fluids promoted by HD may be beneficial in improving cerebral compliance of patients with ESRD. Nevertheless, HD is associated with frequent complications, some of them involve the increased intracranial pressure, such as DDS [6].

Intradialytic hypotension is the commonest complication among HD patients [16–20] and may precede DDS. The symptoms like headache, nausea, and muscle cramps experienced by some patients of the current study during follow-up may represent a milder but not diagnosed spectrum of DDS [5]. Our understanding of the pathophysiology of DDS has improved since its initial description and it is now evident from animal and human studies that DDS is associated with the development of cerebral edema and increased ICP [21,22].

At first DDS was believed to occur only in patients with acute kidney injury when hemodialysis was first initiated, but it has also been reported in patients with CKD [7,23].

According to Castro (2001), DDS can be prevented in patients with very high plasma urea levels by performing low-efficiency dialysis sessions of brief duration, reducing the interdialytic interval, and adding hypertonic solutions such as mannitol in the dialysate, which contribute to reduce cerebral edema [24].

In the current study, we showed that noninvasive ICP parameters (TTP and P2/P1 ratio) were higher in the first HD session of the week (Fig 3). Foley et al. (2011) found that in patients receiving maintenance HD, adverse events including all-cause mortality, myocardial infarction, and hospital admissions occurred more frequently on the day after the long interdialytic interval (1st weekly session) than on other days. We believe that this long (two-day) interdialytic interval contributes to higher interdialytic weight gain (IDWG) and consequently, changes in ICP [25].

Studies using transcranial doppler ultrasonography have evaluated the effects of HD on cerebral hemodynamics and concluded that there is a decrease in CBF after the procedure [26,27]. CBF can influence ICP, just as ICP can influence CBF. An elevated blood flow triggers a response of the self-regulation mechanism of the cerebral circulation that promotes vasodilation, increasing the cerebral blood volume, which consequently increases the ICP. In contrast, when the ICP rises, for any other reason, the cerebral perfusion pressure decreases, which hinders blood circulation and reduces CBF [28].

Considering the role of MAP in brain self-regulation, as it directly influences CBF [29], we evaluated this parameter of the volunteers in this study (Fig 4), however, there was no correlation between MAP and TTP and the P2/P1 ratio. However, the data is interesting because it shows that, in the 1st weekly HD session, MAP is higher, in relation to the 2nd and 3rd session of the week. This may have occurred, due to the greater fluid overload that occurs in the HD session two days' time gap It has also been observed that PAM data, as well as PIC data, are higher in the pre-dialysis moment, in comparison with the post-dialysis.

IDWG is the result of salt and water intake between two HD sessions and is influenced by several factors. It is recommended that IDWG does not exceed 4.5% of 'dry body weight' [30]. Recently, it has been reported that patients with IDWG $\geq$ 5.7% and 4%, respectively, are at an elevated risk for mortality and increased risk for fluid-overload hospitalization [31].

The volunteers in this study had significantly higher IDWG in the first HD session of the week (Fig 5). We hypothesize that patients with an IDWG $\geq$ 4% have worse brain compliance and greater chances of complications, because ICP is derived from cerebral blood and parenchyma and cerebrospinal fluid (CSF) circulatory dynamics, an increase in any of these components (blood, CSF, or parenchyma) may increase ICP [32,33], however, when correlating the

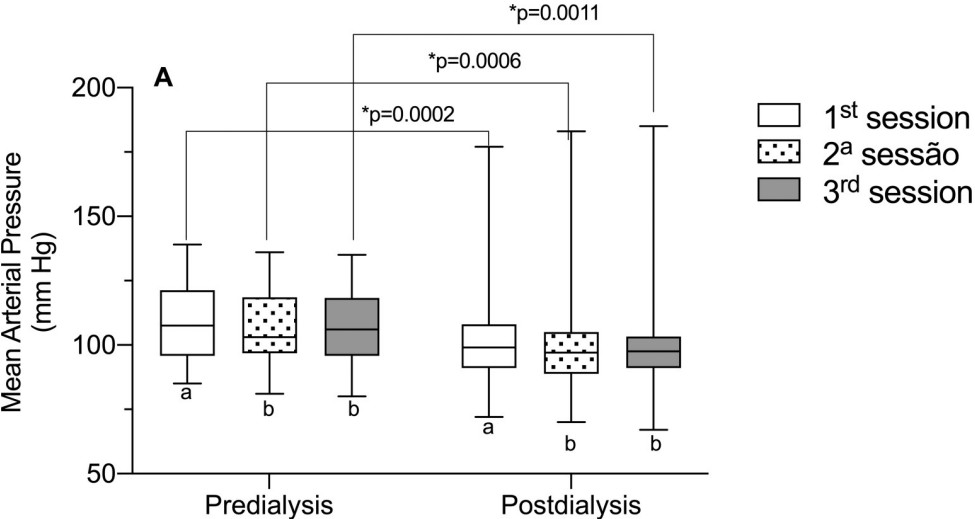

**Fig 4. Mean arterial pressure in the different hemodialysis sessions of the week.** Significant statistical differences were observed for the mean arterial pressure (MAP) parameter when comparing it with the different hemodialysis sessions of the week (1st, 2nd, and 3rd). Different letters, significant differences between the different sessions in the pre- and post-dialysis moments (p<0.05, ANOVA for repeated measures and Tukey's post-test).

PIC parameters (TTP and P2/P1 ratio) with the IDWG, no significant correlation was observed (p>0.05).

A higher IDWG is associated with complications including higher predialysis blood pressure [34,35], intradialytic hypotension as a result of rapid fluid removal during the HD session [36], and increased mortality [37,38], and it may also be related to changes in ICP. Ipema et al. (2016) highlighted the importance of personalized advice on fluid and sodium restriction in HD patients [39].

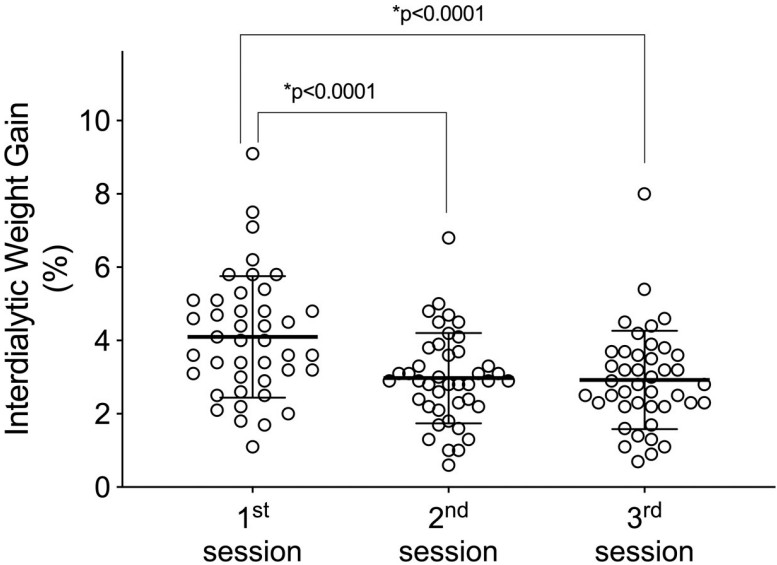

**Fig 5. Interdialytic weight gain in the different hemodialysis sessions of the week.** Significant statistical differences were observed for the interdialytic weight gain parameter when comparing it with the different hemodialysis sessions of the week (1st, 2nd, and 3rd) (p <0.05, ANOVA for repeated measures and Tukey's post-test).

This work was able to demonstrate through a non-invasive method that changes in the ICP of patients undergoing hemodialysis occur, and that these changes are repeated over the months. It is suggested that hemodialysis can improve the parameters of ICP that reflect brain compliance, however, future studies are warranted that examine the causes of ICP alterations, especially considering that prolonged ICP elevation is associated with poor neurocognitive outcomes [40]. Besides that, we show that the noninvasive ICP parameters TTP and P2/P1 ratio were higher in the first weekly HD session than in the second and third sessions, which may happen as a function of the time gap between the last and the first session of the week, which results in a greater accumulation of liquids (Fig 5).

As previously exposed, the noninvasive method used in this study was validated by comparison with the invasive ICP monitoring method [8,10] and has been used in the study of several situations, physiological and pathological, involving the central nervous system [41–45]. Based on this context, the routine uses of non-invasive ICP monitoring in RRT centers could contribute to the clinical evaluation of patients with ESRD.

## Conclusions

Through this unprecedented study, using a non-invasive method, it was possible to understand how the ICP of patients with ESRD behaves. Results of ICP wave morphology analysis of patients with ESRD followed-up for six months by noninvasive ICP monitoring revealed significant differences between predialysis and postdialysis ICP parameters. Also, the pattern of ICP alterations was consistent throughout the study.

### Limitations of the study

Due to the dynamics used to carry out this study, it was not possible to detect complications and correlate them to the ICP. New studies that evaluate patients individually and for a longer time in each dialysis session could explain this issue.

## Supporting information

**S1 File. Average TTP values and P2/P1 ratio in the six-month follow-up.** Mean values of TTP values and P2/P1 ratio in the six months of follow-up, pre- and post-dialysis, of the 42 volunteers included in the study.
(XLSX)

## Author Contributions

**Conceptualization:** Cristiane Rickli, Gustavo Henrique Frigieri, Sérgio Mascarenhas, José Carlos Rebuglio Vellosa.

**Data curation:** Fábio André dos Santos.

**Formal analysis:** Fábio André dos Santos.

**Funding acquisition:** Cristiane Rickli.

**Investigation:** Cristiane Rickli, Lais Daiene Cosmoski, José Carlos Rebuglio Vellosa.

**Methodology:** Cristiane Rickli, Lais Daiene Cosmoski, Fábio André dos Santos, Gustavo Henrique Frigieri, Nicollas Nunes Rabelo, Adriana Menegat Schuinski, Sérgio Mascarenhas.

**Project administration:** Cristiane Rickli, José Carlos Rebuglio Vellosa.

**Resources:** Cristiane Rickli, José Carlos Rebuglio Vellosa.

**Software:** Gustavo Henrique Frigieri, Sérgio Mascarenhas.

**Supervision:** Cristiane Rickli, José Carlos Rebuglio Vellosa.

**Validation:** Cristiane Rickli, Fábio André dos Santos.

**Visualization:** Cristiane Rickli, Gustavo Henrique Frigieri, Nicollas Nunes Rabelo, Adriana Menegat Schuinski, Sérgio Mascarenhas, José Carlos Rebuglio Vellosa.

**Writing – original draft:** Cristiane Rickli, José Carlos Rebuglio Vellosa.

**Writing – review & editing:** Cristiane Rickli, Lais Daiene Cosmoski, Fábio André dos Santos, Gustavo Henrique Frigieri, Nicollas Nunes Rabelo, Adriana Menegat Schuinski, Sérgio Mascarenhas, José Carlos Rebuglio Vellosa.

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
