## [Decision Letter · Decision Letter 0]

12 Jan 2021

PONE-D-20-29529

Use of non-invasive intracranial pressure pulse waveform to monitor patients with End-Stage Renal Disease (ESRD)

PLOS ONE

Dear Professor Vellosa,

Thank you for submitting your manuscript to PLOS ONE. After careful consideration, we feel that it has merit but does not fully meet PLOS ONE’s publication criteria as it currently stands. Therefore, we invite you to submit a revised version of the manuscript that addresses the points raised during the review process.

We look forward to receiving your revised manuscript.

Kind regards,

Patrick Barry Mark

Academic Editor

PLOS ONE

Journal Requirements:

2.Thank you for stating the following in the Competing Interests section:

"I have read the journal's policy and the authors of this manuscript have the following competing interests:

•    CR declares no competing interests/ has nothing to disclose.

•    LDC declares no competing interests/ has nothing to disclose.

•    FAS declares no competing interests/ has nothing to disclose.

•    GHF declares personal fees as employee (Research Coordinator) from Braincare Desenvolvimento e Inovação Tecnológica S.A., during the conduct of the study; In addition, GHF has a patent US9826934B2 issued, and a patent US9993170B1 issued.

•    NNR declares personal fees as medical consultant from Braincare Desenvolvimento e Inovação Tecnológica S.A., during the conduct of the study

•    AMS declares no competing interests/ has nothing to disclose.

•    SM declares he has a patent US9826934B2 issued, and a patent US9993170B1 issued.

•    JCRV declares no competing interests/ has nothing to disclose."

We note that one or more of the authors are employed by a commercial company: Braincare Desenvolvimento e Inovação Tecnológica S.A.

c) We note that you have a patent relating to material pertinent to this article. Please provide an amended statement of Competing Interests to declare this patent (with details including name and number), along with any other relevant declarations relating to employment, consultancy, patents, products in development or modified products etc.

Additional Editor Comments:

This is an interesting paper highlighting a novel technique for monitoring intracranial pressure in dialysis. Overall I thought it was well done and whilst there was some divergence of opinion in the reviewers, I thought it would merit revision and look forward to have a looking at another version.

Reviewers' comments:

Reviewer's Responses to Questions

**Comments to the Author**

1. Is the manuscript technically sound, and do the data support the conclusions?

Reviewer #1: Yes

Reviewer #2: No

2. Has the statistical analysis been performed appropriately and rigorously? 

Reviewer #1: Yes

Reviewer #2: No

3. Have the authors made all data underlying the findings in their manuscript fully available?

Reviewer #1: No

Reviewer #2: Yes

4. Is the manuscript presented in an intelligible fashion and written in standard English?

Reviewer #1: Yes

Reviewer #2: Yes

5. Review Comments to the Author

Reviewer #1: In the manuscript, authors present data from a study assessing the “Use of non-invasive intracranial pressure pulse waveform to monitor patients with End-Stage Renal Disease”. The study is in general of interest and provides some new insights, especially concerning the application of a non-invasive device for ICP measurement. Nevertheless, there are some open questions and thus will not provide full impact for the readership:

Minor issues:

• The data are nicely presented and, I guess, “behave” as expected. Seen differences are like data known for "normal" blood pressure. So, it may reflect just the same phenomenon (removal of fluid) as well? Blood pressure is increasing from end of dialysis session till next start (thus higher at beginning of week). This is all nicely described in the discussion as well. Do you have data on ultrafiltration volume and intradialytic weight gain? These data would be of interest to be included in your analyses. Rate of change from pre- to post-dialytic session could be directly related to UFV and IDWG. Do you have blood pressure data as well? Would be interesting to see the “same” behaviour in these data as well. Please comment.

• Methods for normality testing: Have you checked normality using QQ-plots (or other means to visualize data) or used any formal tests? Just reporting checking of skewness and kurtosis is in my opinion not common in the medical domain (although might be correct).

• Table 1: Are you sure all data are normally distributed? E.g. Are you sure mean HD session length is normally distributed, if 52.4% of the sessions are longer than 240 minutes and the mean is 220 mins? What about the mean HD time? If looking at the SD, I assume they are not normally distributed, thus use of mean and SD are not correct.

• Figure 2 and Figure 3: Are you sure the captions are correct? In Figure 2, I cannot see any comparison of sessions. In Figure 3, I cannot see any data “over the six months of FU”. Furthermore, in Figure 3 it says, “ANOVA is used for comparing data over six months” Please comment.

• Figure 3B, is it correct that for post-dialysis, there are no significant differences? Especially between session 1 and session 2, the difference seems to be “large”.

• In the discussion, you mention that TTP and P2/P1 ratio were higher in the first session compared to the second and third. You argue that it “may happen as a function of the time gap between the last and the first session of the week.” I guess it is not the time, but the fluid overload, thus IDWG.

Reviewer #2: In this study the authors non-invasively recorded intracranial pressure in patients with end-stage renal disease (ESRD) receiving hemodialysis by using the Brain4care device. The authors concluded that intracranial pressure parameters (time to peak and P1/P2) were higher before dialysis compared to after dialysis and that these differences were significantly elevated in the first session relative to the subsequent dialysis sessions. The authors indicated that the rationale for doing this study is based on the fact that some ESRD patients on dialysis may progress into dialysis disequilibrium syndrome suffering of cerebral edema and high intracranial pressure. Although this phenomenon is rare in clinical practice, the authors suggest that the presence of cramps, headaches, fatigue and inability to concentrate after dialysis may be mild manifestations of the dialysis disequilibrium syndrome.

This reviewer has several concerns and comments with this investigation:

1) Although the concept of this work is interesting, I am concerned with the validation of the non-invasive device (Brain4care®) used for measuring intracranial pressure in humans. The authors report that this device measures volumetric changes of the skull in adults by simply applying the sensor through a plastic band around the head and apparently, there is no need for calibration. This appears to be a very simple device in its utilization, but its validation is a concern, particularly when most studies of validation were performed in animal models. Moreover, it seems that the two studies that have addressed validation have been performed by the same group and it appears that there has not been any further independent validation. In addition, data on the device reliability and accuracy in humans is not provided and in consequence, extrapolating a relationship between this device and any gold standard from animals into humans without a complete assessment of reliability may be an important source of study bias.

2) The description of the methodology is succinct and there is no information on the technical details of the measurements. Also, it is not clear what the 4881 data-points from the 42 subjects represent. From this number, I speculate that 116 data points were obtained per subject along the 6-month period at a rate of 3 times sessions per week. Then, we should expect that only 1.61 data-points per patient and session were accomplished. Is the rate of these data points per patient and session sufficient and reliable to provide confidence in these measurements? Unfortunately, without any clear assessment of the reproducibility of the technique any interpretation becomes speculative.

3) It is not clear to this reviewer what would be the contribution of extracranial sources to these measurements and whether this could be a source of artifacts to the readings.

4) Few investigations have used transcranial Doppler from the middle cerebral artery to determine the effects of dialysis on the cerebral circulation. These studies have reported that mean flow velocities remain high before dialysis, decline significantly during dialysis, and stay lower in the post-dialysis period. What is interesting from these studies is that the reduction in flow velocity negatively correlates with the ultrafiltrate volumes, amount of fluid removed and the loss of weight after hemodialysis. Moreover, the post-dialysis reduction in mean flow velocity after 12 months of continued dialysis correlated significantly with the patients’ lower global and executive functions and with progression of their white matter hyperintensities with MRI. It seems then, that these studies are opposite the conclusions achieved by the current study which indicates that intracranial pressure remains high before dialysis. In my opinion, these hemodynamic changes of cerebral blood flow before, during and after dialysis should be discussed in the context of the authors’ findings. [See Stroke 1994;25:408-412; J Am Soc Nephrol 2019;30:147-158.

5) Figure 2 shows the statistical significance of 18 paired comparisons between pre-dialysis and post-dialysis through the use of individual paired student t-tests. In my opinion, these multiple comparisons require an adjustment in the alpha value due to the number of comparisons. Statistical advice is suggested. In addition, Figure 3 displays the comparisons of all measurements through the 6-month period and the authors conclude that session 1 was significantly higher than 2 and 3. Unfortunately, with the large standard deviations and without information on the variability of the measurements, it is difficult to believe that there was a significant difference.

6) An important parameter that would help to understand these changes is systemic blood pressure. However, there was no attempt to document this information. Data on blood pressure may be particularly important as they reported that 64% of the patients were classified with systemic arterial hypertension.

7) In my opinion, most of these hemodynamic changes occur during the dialysis session. If such device is demonstrated to be a valid surrogate of the intracranial pressure, intra-dialytic recordings would be more interesting in order to determine the impact of dialysis on the brain dynamics.

6. PLOS authors have the option to publish the peer review history of their article (what does this mean?). If published, this will include your full peer review and any attached files.

Reviewer #1: No

Reviewer #2: No

---

## [Author Response · Author response to Decision Letter 0]

17 May 2021

Title: Use of non-invasive intracranial pressure pulse waveform to monitor patients with End-Stage Renal Disease (ESRD)

Patrick Barry Mark

Academic Editor

PLOS ONE

Answers to Reviewer Comments

Dear reviewers,

Thank you so much for the valuable comments and critics. Your considerations have certainly provided us with a valuable opportunity to improve our work. Below, you will find an itemized response to each of your concerns:

Reviewer 1:

• The data are nicely presented and, I guess, “behave” as expected. Seen differences are like data known for "normal" blood pressure. So, it may reflect just the same phenomenon (removal of fluid) as well? Blood pressure is increasing from end of dialysis session till next start (thus higher at beginning of week). This is all nicely described in the discussion as well. Do you have data on ultrafiltration volume and intradialytic weight gain? These data would be of interest to be included in your analyses. Rate of change from pre- to post-dialytic session could be directly related to UFV and IDWG. Do you have blood pressure data as well? Would be interesting to see the “same” behaviour in these data as well. Please comment.

Dear reviewer, we value your suggestion and include data on blood pressure and IDWG. The data were included in the manuscript as figure 4 and figure 5. Ultrafiltration data were not obtained. The data on blood pressure and IDWG were correlated to the parameters of ICP (TTP and P2/P1 ratio), however, there was no significant correlation (Person’s correlation).

• Methods for normality testing: Have you checked normality using QQ-plots (or other means to visualize data) or used any formal tests? Just reporting checking of skewness and kurtosis is in my opinion not common in the medical domain (although might be correct).

Dear reviewer, we appreciate your suggestion, and we have modified the method for evaluating the normality of the data. We apply a more formal test (Kolmogorov-Smirnov).

The information has been changed in the manuscript (Material and Methods section).

Below you can see the table with the details of the normality tests (not included in the manuscript).

It can be seen that the points are very approximate to the straight line, so we can assume that the data have distributions that are similar to a normal curve.

The normality of data can be analyzed descriptively through histograms, box-plots, Q-Q Plots, and skewness and kurtosis coefficients, respectively, the degree of deviation or skewness from the symmetry and flatness of the distribution. In addition to descriptive methods, hypothesis tests assess normality, such as the Kolmogorov-Smirnov and Shapiro-Wilks tests. However, it is essential to note that these tests are extremely rigorous and easily reject the hypothesis of normality. Therefore, we should be cautious and not base our decision only on these tests' descriptive levels (p-values). There is no simple relationship between relative power and sample size and no clear rationale for the frequently cited threshold of 30 – 50 patients per group, indicating acceptability of parametric statistics (Vickers 2005).

1. Vickers AJ. Parametric versus non-parametric statistics in the analysis of randomized trials with non-normally distributed data. BMC Med Res Methodol. 2005 Nov 3;5:35. doi: 10.1186/1471-2288-5-35. PMID: 16269081; PMCID: PMC1310536.

• Table 1: Are you sure all data are normally distributed? E.g. Are you sure mean HD session length is normally distributed, if 52.4% of the sessions are longer than 240 minutes and the mean is 220 mins? What about the mean HD time? If looking at the SD, I assume they are not normally distributed, thus use of mean and SD are not correct.

Dear reviewer, Table 1 presents the data concerning the subjects' characteristics included in this clinical trial. We did not perform inferential statistics with the data shown in Table 1. Therefore, the normality of the data presented in Table 1 does not influence our study's results and conclusions.

• Figure 2 and Figure 3: Are you sure the captions are correct? In Figure 2, I cannot see any comparison of sessions. In Figure 3, I cannot see any data “over the six months of FU.” Furthermore, in Figure 3 it says, “ANOVA is used for comparing data over six months” Please comment.

In figure 2, we consider the two parameters (A. TTP and B. P2/P1 ratio); We do not include statistical significance values in the figure since we did not find significant differences between the evaluation periods. Therefore the indication of the statistic in the face of non-significant differences is not necessary. However, if the Reviewer/Editor considers this information relevant, we can include it in the manuscript.

In figure 3, we show the data regarding the follow-up during the six months of the study. For the data analysis, the means were obtained for the first, second, and third hemodialysis sessions during the six months of the research. Thus, the data from the sessions were obtained during the entire follow-up period (six months).

Analysis of Variance (ANOVA) is a common and robust statistical test that you can use to compare the mean scores collected from different conditions or groups in an experiment. 

A repeated-measures (or within-participants) test is what you use when you want to compare the performance of the same group of participants in different experimental moments. That is, when the same participants take part in all of the conditions in your study.

• Figure 3B, is it correct that for post-dialysis, there are no significant differences? Especially between session 1 and session 2, the difference seems to be “large”.

The reviewer has raised an important point in Figure 3B. We have checked the data, and there are no significant differences between the predialysis and post-dialysis times. Figure 3B the y-axis shows a maximum value for P2/P1 ratio of 1.7, and the lower limit of the graph starts at 0.5. This way, the visual difference in the figure becomes more evident. Therefore, Figure 3B was re-edited with the y-axis starting at zero.

• In the discussion, you mention that TTP and P2/P1 ratio were higher in the first session compared to the second and third. You argue that it “may happen as a function of the time gap between the last and the first session of the week.” I guess it is not the time, but the fluid overload, thus IDWG.

Dear reviewer, your note is correct and that is what we wanted to suggest. The sentence was rewritten in the manuscript.

Reviewer 2:

1) Although the concept of this work is interesting, I am concerned with the validation of the non-invasive device (Brain4care®) used for measuring intracranial pressure in humans. The authors report that this device measures volumetric changes of the skull in adults by simply applying the sensor through a plastic band around the head and apparently, there is no need for calibration. This appears to be a very simple device in its utilization, but its validation is a concern, particularly when most studies of validation were performed in animal models. Moreover, it seems that the two studies that have addressed validation have been performed by the same group and it appears that there has not been any further independent validation. In addition, data on the device reliability and accuracy in humans is not provided and in consequence, extrapolating a relationship between this device and any gold standard from animals into humans without a complete assessment of reliability may be an important source of study bias.

Dear reviewer, in the discussion session, we cite studies that assessed non-invasive ICP in humans by the method used in this study, such as Ballestero MFM, Frigieri G, Cabella BCT, de Oliveira SM, de Oliveira RS. Prediction of intracranial hypertension through noninvasive intracranial pressure waveform analysis in pediatric hydrocephalus. Child’s Nerv Syst. 2017; 33: 1517–1524. In addition, the method in question is registered with the National Health Surveillance Agency (registration number 81157910004) and is present in reference hospitals such as Hospital Sírio Libanês, Hospital Nove de Julho, Beneficiência Portuguesa – BP and Neuresp Neurologia Especializada. This study did not aim to correlate the Brain4care method with the gold standard, not least because in practice the invasive method is not routinely used in renal patients. The method used in this study provides a unique strategy for evaluating patients who do not have any information on brain compliance.

2) The description of the methodology is succinct and there is no information on the technical details of the measurements. Also, it is not clear what the 4881 data-points from the 42 subjects represent. From this number, I speculate that 116 data points were obtained per subject along the 6-month period at a rate of 3 times sessions per week. Then, we should expect that only 1.61 data-points per patient and session were accomplished. Is the rate of these data points per patient and session sufficient and reliable to provide confidence in these measurements? Unfortunately, without any clear assessment of the reproducibility of the technique any interpretation becomes speculative.

To work with the large volume of data collected, the TTP values and the P2 / P1 ratio of the different sessions (1st, 2nd and 3rd) were initially averaged for each month of follow-up (file sent in .xlm format). In addition, within the 6 months that each volunteer was followed up, there were eventually days when the volunteer's brain compliance was not monitored (due to not accepting on a certain day, not feeling well or missing the hemodialysis session).

3) It is not clear to this reviewer what would be the contribution of extracranial sources to these measurements and whether this could be a source of artifacts to the readings.

Dear reviewer, the extracranial sources that could generate artifacts refer to the patient's posture (sitting, lying or standing) and movement during monitoring. As for posture, patients were always monitored in the same way in the pre- and post-dialysis moments, sitting on their own hemodialysis chairs, in addition, the sensor was placed on the same side of the head at both times (pre- and post-dialysis). dialysis). Regarding the patient's movement during monitoring, the volunteers were instructed to remain immobile during the procedure, however, if movement occurred, the acquisition was totally modified and visible, as it is an extremely sensitive method that captures cranial microalterations. In this sense, the period that the patient moved was removed from the analysis and the volunteer was monitored for a longer time to compensate for the excluded period.

4) Few investigations have used transcranial Doppler from the middle cerebral artery to determine the effects of dialysis on the cerebral circulation. These studies have reported that mean flow velocities remain high before dialysis, decline significantly during dialysis, and stay lower in the post-dialysis period. What is interesting from these studies is that the reduction in flow velocity negatively correlates with the ultrafiltrate volumes, amount of fluid removed and the loss of weight after hemodialysis. Moreover, the post-dialysis reduction in mean flow velocity after 12 months of continued dialysis correlated significantly with the patients’ lower global and executive functions and with progression of their white matter hyperintensities with MRI. It seems then, that these studies are opposite the conclusions achieved by the current study which indicates that intracranial pressure remains high before dialysis. In my opinion, these hemodynamic changes of cerebral blood flow before, during and after dialysis should be discussed in the context of the authors’ findings. [See Stroke 1994;25:408-412; J Am Soc Nephrol 2019;30:147-158.

We would like to thank you the suggestion, we adjusted the text in the discussion session.

5) Figure 2 shows the statistical significance of 18 paired comparisons between predialysis and post-dialysis through the use of individual paired student t-tests. In my opinion, these multiple comparisons require an adjustment in the alpha value due to the number of comparisons. Statistical advice is suggested. In addition, Figure 3 displays the comparisons of all measurements through the 6-month period and the authors conclude that session 1 was significantly higher than 2 and 3. Unfortunately, with the large standard deviations and without information on the variability of the measurements, it is difficult to believe that there was a significant difference.

In Figure 2, the paired t-test was applied considering predialysis and postdialysis in each evaluated period. Thus, for each pairwise comparison, the use of the paired t-test is adequate. We did not perform multiple comparisons with the paired t-test. The Bonferroni correction for alpha adjustment would be applied if multiple comparisons were made. A second analysis involving the different periods within the same group; we employed the ANOVA test for repeated measures for this purpose.

The reviewer has raised an important point in Figure 3B. We have checked the data, and there are no significant differences between the predialysis and post-dialysis times. Figure 3B the y-axis shows a maximum value for P2/P1 ratio of 1.7, and the lower limit of the graph starts at 0.5. This way, the visual difference in the figure becomes more evident. Therefore, Figure 3B was re-edited with the y-axis starting at zero.

In Table 2, we can see the distribution of the data. The descriptive statistics show a low coefficient of variation. The mean and median values are very similar. The standard deviation and interquartile range present low values. The skewness and kurtosis coefficients are between -1 and +1, indicating that the data demonstrate an approximately normal distribution.

In figure 2 (below), we can see the frequency histograms and Q-Q Plot. The figure supports the normal distribution of the analyzed data. The histograms show a distribution very similar to a normal distribution. The Q-Q Plot has the vast majority of points fitted to a straight line, indicating a normal distribution approximation.

Therefore, this information reinforces the reliability of the statistical analysis performed.

In Table 3, we can see the distribution of the data from the P2/P1 ratio. The results are similar to those observed in Table 2. The descriptive statistics show a low coefficient of variation. The mean and median values are very similar. The standard deviation and interquartile range present low values. The skewness and kurtosis coefficients are between -1 and +1, indicating that the data suggest an approximately normal distribution.

In figure 3 (below), we can see the frequency histograms and Q-Q Plot. The figure supports the normal distribution of the analyzed data. The histograms show a distribution very similar to a normal distribution. The Q-Q Plot has the vast majority of points fitted to a straight line, indicating a normal distribution approximation.

Therefore, this information reinforces the reliability of the statistical analysis performed.

6) An important parameter that would help to understand these changes is systemic blood pressure. However, there was no attempt to document this information. Data on blood pressure may be particularly important as they reported that 64% of the patients were classified with systemic arterial hypertension.

Dear reviewer, we value your suggestion and include data on blood pressure. The data were included in the manuscript as figure 4. The data on blood pressure were correlated to the parameters of ICP (TTP and P2/P1 ratio), however, there was no significant correlation (Person’s correlation).

7) In my opinion, most of these hemodynamic changes occur during the dialysis session. If such device is demonstrated to be a valid surrogate of the intracranial pressure, intra-dialytic recordings would be more interesting in order to determine the impact of dialysis on the brain dynamics.

Dear reviewer, your collocations are interesting, however, in this study the objective was to evaluate the effect of hemodialysis on cerebral compliance and for that, the patients were monitored for a few minutes before and after dialysis. In this study, it was not feasible to monitor the patient during part or all the hemodialysis session for the following reasons: 42 volunteers were monitored for a period of 6 months; we believe that we would have patients drop out if, in order to participate in the study, they had to be monitored in all their sessions for a "long" period. In this study, it was not feasible to keep a patient connected to the equipment for a long period, due to the number of equipment and researchers available to carry out the evaluations. However, this study is the first to clearly demonstrate the effect of hemodialysis on brain compliance and most importantly, it was not occasional changes, but reproducible over the months. It is expected and positive that the results of this study will awaken many other questions to be answered. Future studies, which continuously monitor hemodialysis sessions of patients with end-stage renal disease and acute kidney disease, could assist in establishing the application of the non-invasive method for assessing intracranial pressure in these situations.

---

## [Editor Report · Decision Letter 1]

27 May 2021

Use of non-invasive intracranial pressure pulse waveform to monitor patients with End-Stage Renal Disease (ESRD)

PONE-D-20-29529R1

Dear Dr. Vellosa,

We’re pleased to inform you that your manuscript has been judged scientifically suitable for publication and will be formally accepted for publication once it meets all outstanding technical requirements.

Kind regards,

Patrick Barry Mark

Academic Editor

PLOS ONE

Additional Editor Comments (optional):

Thank you for revising this interesting study
---

## [Editor Report · Acceptance letter]

14 Jul 2021

PONE-D-20-29529R1 

Use of non-invasive intracranial pressure pulse waveform to monitor patients with End-Stage Renal Disease (ESRD) 

Dear Dr. Vellosa:

I'm pleased to inform you that your manuscript has been deemed suitable for publication in PLOS ONE. Congratulations! Your manuscript is now with our production department. 

Kind regards, 

on behalf of

Prof Patrick Barry Mark 

Academic Editor

PLOS ONE